# Effect of Vitamin D Supplementation on Survival of Digestive Tract Cancer Patients with Low Bioavailable 25-Hydroxyvitamin D Levels: A Post Hoc Analysis of the AMATERASU Randomized Clinical Trial

**DOI:** 10.3390/cancers12020347

**Published:** 2020-02-04

**Authors:** Mitsuyoshi Urashima, Mai Okuyama, Taisuke Akutsu, Hironori Ohdaira, Mutsumi Kaji, Yutaka Suzuki

**Affiliations:** 1Division of Molecular Epidemiology, Jikei University School of Medicine, 3-25-8, Nishi-Shimbashi, Minato-Ku, Tokyo 105-8461, Japan; maiokuyama0511@gmail.com (M.O.); taisuke0107.jusom@gmail.com (T.A.); 2Department of Surgery, International University of Health and Welfare Hospital, 537-3 Iguchi, Nasushiobara, Tochigi 329-2763, Japan; ohdaira@iuhw.ac.jp (H.O.); m.kaji0528@gmail.com (M.K.); yutaka@iuhw.ac.jp (Y.S.)

**Keywords:** 25-hydroxyvitamin D, total 25(OH)D, bioavailable 25(OH)D, vitamin D, cancer mortality, overall survival, relapse-free survival, colorectal cancer, gastric cancer, esophageal cancer

## Abstract

Vitamin D has been shown to suppress the growth of cancer cells. Cancer cells are believed to take up bioavailable 25-hydroxyvitamin D (25[OH]D) (i.e., not bound to vitamin-D-binding protein (DBP)) more efficiently than DBP-bound 25(OH)D. Our aim was to use this bioavailable 25(OH)D, rather than total 25(OH)D, as a biomarker of vitamin D deficiency to investigate whether vitamin D supplementation improves the relapse-free survival (RFS) of patients with digestive tract cancer from the esophagus to the rectum by conducting a post hoc analysis of the AMATERASU trial (UMIN000001977). The bioavailable 25(OH)D levels were calculated via an equation using data of serum total 25(OH)D, albumin, and DBP levels, and DBP genotypes (rs7041 and rs4588). We estimated bioavailable 25(OH) levels in 355 patients. In a subgroup of patients with low bioavailable 25(OH)D levels (<median) (*n* = 177), 5 year RFS was 77% in the vitamin D group vs. 58% in the placebo group (hazard ratio, 0.54; 95% confidence interval, 0.31–0.95; *p* = 0.03), whereas no significant difference was seen in a subgroup of patients with high bioavailable 25(OH)D levels (*p* for interaction = 0.046). We hypothesize that vitamin D supplementation may be effective in improving RFS among digestive tract cancer patients with low bioavailable 25(OH)D levels.

## 1. Introduction

Vitamin D can be obtained from skin exposed to sunlight, diet, or supplements. It is metabolized in the liver to 25-hydroxyvitamin D (25[OH]D), and is further activated in the kidneys through 1α−hydroxylase to 1,25-dihydroxyvitamin D (1,25[OH]2D), which is considered to facilitate calcium absorption and improve bone health [1]. 1,25[OH]2D is metabolized by the enzyme CYP24A1, limiting calcitriol action by catabolism. Renal CYP27B1 gene expression is regulated and activated by the parathyroid hormone. Following synthesis, 1,25[OH]2D is released into the serum; in addition to its action on bone, it can also act on the intestine and kidney to regulate calcium metabolism. CYP27B1 and CYP24A1 are found in numerous tissues throughout the body, including the skin, colon, pancreas, liver, brain, and placenta, allowing for 1,25[OH]2D synthesis and degradation [2]. In addition to this classical pathway, the 25(OH)D molecule in the serum has been hypothesized to be absorbed by cancer cells, to be activated by 1α−hydroxylase and bind to vitamin D nuclear receptors within the cell, to inhibit aberrant cellular proliferation by regulating a variety of genes, and to prevent cancer relapse [3]. A meta-analysis of 16 prospective cohort studies including more than 100,000 patients showed that higher total 25(OH)D levels in serum were significantly associated with decreased cancer-specific mortality [4] independent of cancer site or pathological subtype. However, contrary to expectations, recent randomized clinical trials (RCTs) have demonstrated that vitamin D supplementation, compared with placebo, did not result in significant improvement of relapse or progression-free survival among the total study population or a subgroup of patients with low total 25(OH)D levels (<20 ng/mL) in digestive tract cancer [5] and advanced or metastatic colorectal cancer [6]. 

Recently, the assessment of vitamin D status has started to shift from examining total 25(OH)D levels to examining bioavailable 25(OH)D levels [7]. Around 90% of total 25(OH)D is carried by vitamin-D-binding protein (DBP), whereas non-DBP-bound 25(OH)D, so called bioavailable 25(OH)D, is mainly carried by albumin, and residual 25(OH)D exists as free form [8]. Because DBP has a 1000-fold stronger affinity for 25(OH)D than albumin, cancer cells cannot easily take up DBP-bound 25(OH)D, but they can easily take up bioavailable 25(OH)D [9]. In a prospective cohort study of 1031 patients with hepatocellular carcinoma, higher bioavailable levels, but not total or free 25OHD levels, were associated with improved survival [10]. We thus hypothesized that bioavailable 25(OH)D might be a more appropriate marker of vitamin D insufficiency than total 25(OH)D levels, and that vitamin D supplementation may improve the survival of patients with low bioavailable levels of 25(OH)D.

We previously conducted the AMATERASU randomized, double-blind, placebo-controlled trial of postoperative oral vitamin D3 supplementation (2000 IU/day) in 417 patients with stage I to III digestive tract cancer from the esophagus to the rectum who underwent curative surgery (UMIN000001977) [5]. In this post hoc analysis using residual serum samples and clinical data from the AMATERASU trial, we aimed to explore whether vitamin D supplementation improved survival in a subgroup of patients with lower than median levels of bioavailable 25(OH)D. 

## 2. Results

### 2.1. Study Population

Table 1 shows the characteristics of the participants analyzed in this post hoc study. The distribution of variables was similar to the original trial [5]. All patients were Japanese. The only significant difference between allocation groups was in age. 

### 2.2. Genetic Polymorphisms of Vitamin-D-Binding Protein

The T allele at rs7041 was observed in 76% of participants and was negatively associated with total 25(OH)D levels (Table 2). The A allele at rs4588 was observed in 29% participants and was also negatively associated with total 25(OH)D levels. Hardy–Weinberg equilibrium tests were not significant for either rs7041 (*p* = 0.33) or rs4588 (*p* = 0.36).

### 2.3. Changes in Albumin, DBP, and Free and Bioavailable 25(OH)D Levels 

In the AMATERASU trial, at a median 23.5 (interquartile range (IQR), 13–44) days after surgery, participants were randomized and began study supplementation. Changes in serum albumin, DBP, free 25(OH)D, and bioavailable 25(OH)D levels were compared between just before taking trial supplements (pre) and 1 year after start of supplements (post). Percent changes within each group were compared using the Wilcoxon signed-rank test, and changes between groups were compared with the Mann–Whitney test (Figure 1). Albumin levels increased significantly 1 year after surgery in both groups, although there were no significant differences between groups (Figure 1A). DBP levels decreased significantly within both groups, but there were no significant differences between groups (Figure 1B). Free and bioavailable 25(OH)D levels increased significantly within both groups, and changes were significantly higher in the vitamin D group than in the placebo group (Figure 1C,D). Median total 25(OH)D levels increased significantly in the vitamin D group but did not change significantly in the placebo group (data previously reported [5]).

### 2.4. Relapse-Free Survival of Subgroups Stratified by Median Bioavailable 25(OH)D Levels 

A total of 355 patients were divided into two subgroups stratified at the median (1.71 ng/mL) bioavailable 25(OH)D level before starting supplementation. The effects of vitamin D supplementation on relapse-free survival (RFS) were then compared between the low and high subgroups (Figure 2). Among patients with low bioavailable 25(OH)D levels (*n* = 177), the 5 year RFS was 77% in the vitamin D group and 58% in the placebo group (hazard ratio (HR) for relapse or death, 0.54; 95% confidence interval (CI), 0.31–0.95) (Figure 2A). In contrast, among patients with high bioavailable 25(OH)D levels (*n* = 178), the RFS did not differ significantly between treatment groups (Figure 2B). There was a significant two-way interaction between the subgroups of low bioavailable 25(OH)D levels and vitamin D supplementation (*p* for interaction = 0.046).

### 2.5. Overall Survival of Subgroups Stratified at Median Bioavailable 25(OH)D Levels

Effects of vitamin D on overall survival (OS) were compared between the low and high bioavailable 25(OH)D subgroups (Figure 3). In the subgroup of patients with low bioavailable 25(OH)D levels, the 5 year OS in the vitamin D group was 84%, which tended to be higher than the 69% recorded in the placebo group, although the difference was not significantly different (Figure 3A). In the subgroup of patients with high bioavailable 25(OH)D levels, the 5 year OS in the vitamin D group vs. placebo group was 79% vs. 90%, respectively, with no significant differences between groups (Figure 3B). There was a significant two-way interaction between the subgroup of low bioavailable 25(OH)D levels and vitamin D supplementation (*p* for interaction = 0.036).

### 2.6. RFS of Subgroups Stratified by Median Free 25(OH)D Level

A total of 355 patients were divided into two subgroups stratified at the median (5.70 pg/mL) free 25(OH)D level before starting supplementation. The effects of vitamin D supplementation were then compared between the subgroups of low and high free 25(OH)D levels (Figure 4). In the subgroup of patients with low free 25(OH)D levels (*n* = 177), the 5 year RFS was 73% in the vitamin D group, which tended to be higher than the 62% recorded in the placebo group, although differences were not significant (Figure 4A). On the other hand, in the subgroup of patients with high free 25(OH)D levels (*n* = 178), the 5 year RFS was almost the same between groups (Figure 4B). There was no significant two-way interaction between the subgroups of low free 25(OH)D and vitamin D supplementation.

### 2.7. OS of Subgroups Stratified by Median Free 25(OH)D Level

The effects of vitamin D on OS were compared between the low and high free 25(OH)D level groups (Figure 5). The 5 year OS in the vitamin D group vs. placebo group was 80% vs. 75% in the subgroup of patients with low free 25(OH)D levels, and 83% vs. 85% in the subgroup of patients with high free 25(OH)D levels, respectively, with no significant differences between groups. Similarly, there was no significant two-way interaction between the subgroup of low free 25(OH)D levels and vitamin D supplementation.

### 2.8. Adverse Events of Bioavailable 25(OH)D

In the subgroup of patients with low bioavailable 25(OH)D levels, fractures occurred in one patient (1.0%) in the vitamin D group vs. five patients (6.6%) in the placebo group (*p* = 0.04). In the subgroup of patients with high bioavailable 25(OH)D levels, fractures occurred in two patients (1.7%) in the vitamin D group and one patient (1.6%) in the placebo group, which was not a significant difference (*p* = 0.94) (Table 3). Frequencies of other adverse events did not differ significantly between groups. 

## 3. Discussion

In this post hoc analysis of the AMATERASU trial, daily supplementation with 2000 IU of vitamin D significantly improved 5 year RFS in the subgroup of patients with low bioavailable 25(OH)D levels, but not in the subgroup with high bioavailable 25(OH)D levels. In our previous analysis of the AMATERASU trial, vitamin D was not effective in the subgroup of patients with low serum levels (<20 ng/mL) of total 25(OH)D at baseline, contrary to our expectation that vitamin D would be effective only in the subgroup with low total 25(OH)D [4]. However, in this post hoc study, we demonstrated that vitamin D supplementation may be effective only in cancer patients with low bioavailable 25(OH)D levels. As expected, our findings thus suggest that the level of bioavailable 25(OH)D, but not the total level of bound 25(OH)D and free 25(OH)D, may be a good biomarker of vitamin D deficiency when vitamin D supplementation is used for cancer patients. Similarly, the 5 year OS was 15% higher in the vitamin D group than in the placebo group among the subgroup of patients with low bioavailable 25(OH)D levels, although the difference was not significant. Moreover, there was a significant interaction between vitamin D supplementation and the subgroup of patients with low bioavailable 25(OH)D levels in HRs of both relapse/death and all-cause death. On the other hand, no such significant differences and interactions were observed in the subgroup of patients with low free 25(OH)D levels. To our knowledge, this is the first report showing that vitamin D supplementation may improve the survival of patients with low bioavailable 25(OH)D levels in patients with digestive tract cancer. Other studies have shown that high bioavailable levels, but neither free nor total 25(OH)D levels, are positively associated with survival time of patients with hepatocellular carcinoma [10] and with colorectal cancer [11], although both studies were observational. Those results are considered consistent with the results of this study. Future confirmatory RCTs should include cancer patients with low bioavailable 25(OH)D levels to confirm these findings. 

Compared with a median 23.5 (IQR, 13–44) days after surgery, albumin levels increased and DBP levels decreased 1 year after starting supplements, independent of treatment group. Although we did not measure these levels before surgery, results suggest that surgical stress may decrease albumin levels, probably via accelerated catabolism, and that DBP levels may increase by unknown mechanisms. In previous studies, 1 to 2 days after surgery, total 25(OH)D levels were reported to decrease to nadir levels and to start recovering 6 days after surgery [12,13]. We hypothesized that the mechanism of increase in DBP occurred through surgical stress and reduced total 25(OH)D levels, which subsequently increased DBP levels through a negative feedback loop; the increased DBP helped total 25(OH)D levels to recover. We hypothesized that the mechanism of increase in DBP occurred through surgical stress and reduced total 25(OH)D levels, which subsequently increased DBP levels through a negative feedback loop. The increased DBP levels aided the recovery of total 25(OH)D levels.

Vitamin D supplementation to decrease the risk of fracture is still inconclusively supported [14,15]. In this study, in the subgroup of patients with low bioavailable 25(OH)D levels, bone fracture was observed significantly less frequently in the vitamin D group than the placebo group, although the sample size was small. Moreover, a prospective cohort study demonstrated that black Americans had lower levels of total 25(OH)D levels and DBP-bound 25(OH)D levels than white Americans, resulting in similar estimated bioavailable 25(OH)D levels [16]. That study may explain, at least in part, the paradox of why black Americans have lower total 25(OH)D levels but higher bone mineral density than white Americans [16]. Thus, vitamin D supplementation may prevent fractures among people with low bioavailable 25(OH)D levels. 

In the present study, the incidence of cancer that appeared de novo in organs other than the site of the primary cancer after starting supplementation did not differ between treatment groups among the subgroup of patients with low bioavailable 25(OH)D levels. A prospective case–control study of advanced prostate cancer cases and controls showed that pre-diagnosis levels of total, but not free or bioavailable 25(OH)D levels, were associated with lower colorectal cancer risk [17,18]. Thus, bioavailable 25(OH)D levels may modify the effects of vitamin D supplementation on survival of cancer patients, but may not impact the risk for cancer incidence. However, identification of the optimal form and dosage of vitamin D supplementation that may provide effective protection against cancer warrants further investigation [19]. 

This trial had several limitations. First, it was a post hoc subgroup analyses of the AMATERASU clinical trial. As such, the sample size was not calculated for subgroup analyses. In addition, subgroup analyses may have had increased probability of a type I error due to multiple comparisons. Thus, the findings must be considered exploratory and interpreted with caution, although the main results of this study were remarkable. Second, the number analyzed was less than in the original study, because four factors (total 25(OH)D, polymorphisms of DBP, serum levels of DBP, and albumin) were needed to calculate bioavailable 25(OH)D levels. However, patient characteristics in this post hoc study were similar to those in the original trial [5]. Third, in the present trial, patients underwent blood sampling to measure serum total 25(OH)D levels at the first outpatient visit between 2 and 6 weeks after surgery, but not preoperatively. Serum levels of 25(OH)D and albumin have been reported to decrease after surgery [12,13]. Thus, baseline levels of total and bioavailable 25(OH)D could have been affected by stress induced by surgery and cannot be exactly compared with 25(OH)D levels in other studies in which blood sampling took place before surgery or before a diagnosis of cancer. Fourth, the allele frequencies of DBP polymorphisms differ among races [16]. The variant allele frequencies of rs7401 and rs4588 in this Japanese study population were 0.76, close to the 0.83 found in black Americans but not close to the 0.42 of white Americans, and 0.29, close to 0.28 in white Americans but not close to the 0.10 of black Americans, respectively [16]. Thus, the results of this Japanese study are not necessarily generalizable to other populations. Fifth, free and bioavailable 25(OH)D levels were based on calculations rather than direct measurements, although these measurements have been confirmed to be well-correlated [16]. 

## 4. Materials and Methods 

### 4.1. Study Design

This study was a post hoc analysis of the AMATERASU trial conducted in Japan. Details of the study have been previously reported [4]. Briefly, 417 patients with digestive tract cancer from the esophagus to the rectum participated in a randomized, double-blind, clinical trial to compare the effects of vitamin D3 supplements (2000 IU/day) and placebo on RFS and OS at an allocation ratio of 3:2, at the International University of Health and Welfare Hospital (Ohtawara, Tochigi prefecture, Japan) between January 2010 and February 2018. The follow-up rate was 99.8%. Serum samples were obtained to measure 25(OH)D concentrations at a median of 23.5 (IQR, 13–44) days postoperatively and approximately 1 year later. The ethics committee of the International University of Health and Welfare Hospital (ethics approval code: 13-B-263) and the Jikei University School of Medicine (ethics approval code: 21-216(6094)) approved the trial protocol. Written, informed consent was obtained from each participating patient before surgery. Details of inclusion and exclusion criteria are described in the original report [5]. Briefly, the trial included patients not taking vitamin D supplements, with stage I to III digestive tract cancer (48% colorectal, 42% gastric, and 10% esophageal) who underwent curative surgery with complete tumor resection. The primary outcome was 5 year RFS, defined as elapsed time from the date of randomization (i.e., time from starting the supplement to the earliest date of cancer relapse or death from any cause). Secondary outcome was 5 year OS, defined as elapsed time from the date of randomization (i.e., time from starting the supplement to the date of death from any cause) in this study.

### 4.2. Bioavailable 25(OH)D Concentrations and Free 25(OH)D Levels

Serum 25(OH)D concentrations were prospectively measured using a radioimmunoassay (SRL Inc., Hachioji, Tokyo, Japan) at the first outpatient visit, i.e., 2 to 6 weeks after surgery, just before starting supplementation, and 1 year later, as described in the original report [5]. Single-nucleotide polymorphisms (SNPs) of DBP1 (rs7041) and DBP2 (rs4588) were prespecified and genotyped as described in the original report [5]. These SNPs are missense mutations that have been previously reported to influence the serum levels of 25(OH)D [20]. Serum samples were stored at −80 °C prior to use. Retrospectively, i.e., after obtaining the main results [5], serum DBP concentrations were measured using residual serum samples and ELISA kits from Abcam (Cambridge, MA, USA) according to the manufacturer’s protocols. The serum samples were diluted 100 times with diluent. Paired concentrations of DBP at baseline (pre) and 1 year after randomization (post) were measured on the same plate. Bioavailable 25(OH)D concentrations were calculated using serum concentrations of 25(OH)D, DBP, albumin, and a combination of DBP genotypes, before and 1 year after starting supplementation, referring to the equations in Powe’s report [16]. Serum albumin data on the same day of total 25(OH)D measurements were retrospectively collected from hospital charts. SNPs of DBP and serum total 25(OH)D levels were measured prospectively during the main study. In this post hoc study, serum DBP levels and serum albumin levels were additionally measured. These four factors were required to calculate the levels of both bioavailable and free 25(OH)D in the 216 subjects (86%) in the vitamin D group and the 139 subjects (84%) in the placebo group included in the final analysis (Figure 6).

### 4.3. Statistical Analysis

All patients who underwent randomization and for whom it was possible to calculate bioavailable 25(OH)D levels were included in this analysis. Dichotomous and continuous variables between the vitamin D and placebo group were compared via chi-squared test and Mann–Whitney test, respectively. Paired and unpaired continuous variables were compared via Wilcoxon signed-rank test and the Mann–Whitney test, respectively. Relapse- and death-related outcomes were assessed according to the randomization group, whether or not supplements were taken. Subgroups were stratified by the median level into either low or high bioavailable 25(OH)D levels and free 25(OH)D levels. The effects of vitamin D and placebo on risk of relapse or death and total death were estimated using Nelson–Aalen cumulative hazard curves for outcomes. A Cox proportional hazards model was used to determine HR and 95% CI. To clarify whether vitamin D supplementation significantly affected these subgroups, *P* for interaction was analyzed based on a Cox regression model including three variables: (1) vitamin D group; (2) low bioavailable 25(OH)D level group; and (3) vitamin D group and low bioavailable 25(OH)D group multiplied together as an interaction variable, by two-way interaction tests comparing low and high subgroups. Values with two-sided *p* < 0.05 were considered significant. All data were analyzed using Stata 14.0 (StataCorp LP, College Station, TX, USA).

## 5. Conclusions

This is the first report suggesting that vitamin D supplementation may be effective in improving RFS among digestive tract cancer patients with low bioavailable 25(OH)D levels. However, because this was a post hoc subgroup analysis, the findings must be considered exploratory and interpreted with caution.

## Figures and Tables

**Figure 1 cancers-12-00347-f001:**
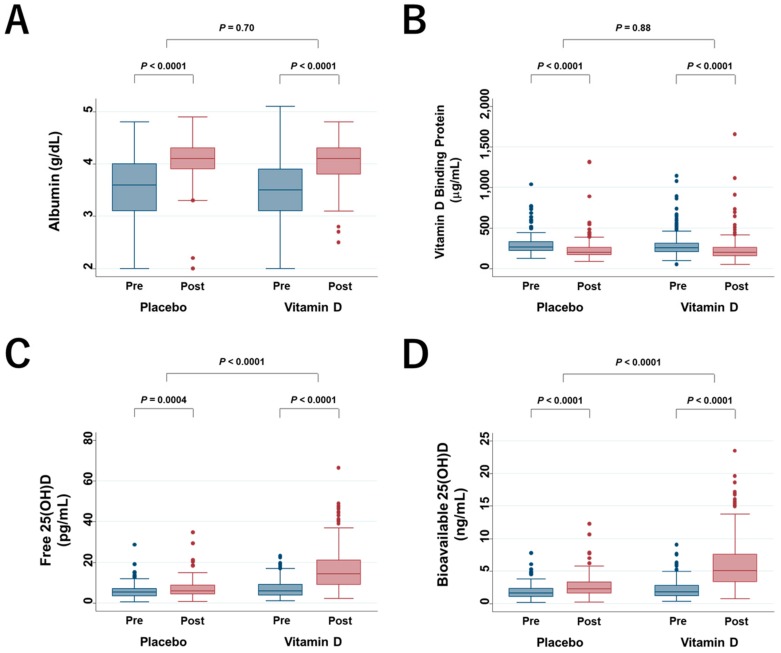
Box plot of changes in serum albumin levels (**A**), vitamin-D-binding protein (**B**), free 25(OH)D (**C**), and bioavailable 25(OH)D (**D**). Pre: just before taking trial supplements; post: 1 year after beginning to take the supplements; 25(OH)D: 25-hydroxyvitamin D.

**Figure 2 cancers-12-00347-f002:**
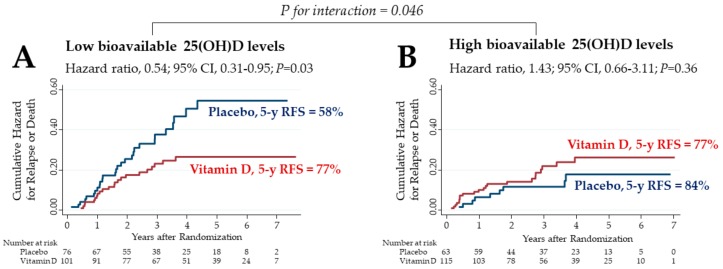
Nelson–Aalen cumulative hazard curves for relapse or death in the subgroups of bioavailable 25(OH)D levels lower (**A**) and higher (**B**) than median levels (1.71 ng/mL) before starting supplementation. 25(OH)D: 25-hydroxyvitamin D; CI: confidence interval; 5 y RFS: 5 year relapse-free survival.

**Figure 3 cancers-12-00347-f003:**
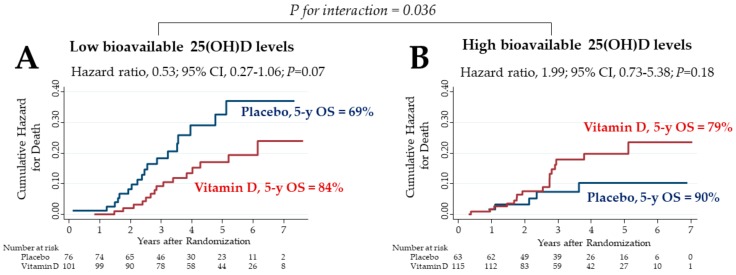
Nelson–Aalen cumulative hazard curves for death in the subgroups of bioavailable 25(OH)D levels lower (**A**) and higher (**B**) than median levels (1.71 ng/mL) before starting supplementation. 25(OH)D: 25-hydroxyvitamin D; CI: confidence interval; 5 y OS: 5 year overall survival.

**Figure 4 cancers-12-00347-f004:**
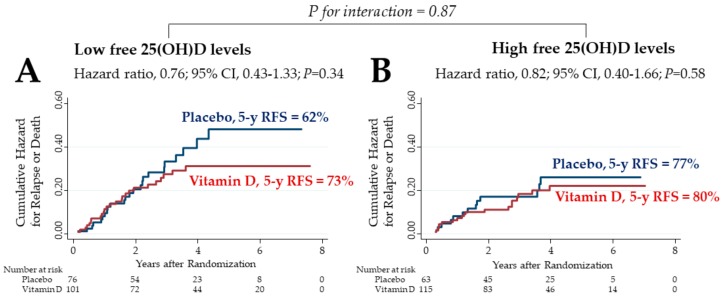
Nelson–Aalen cumulative hazard curves for relapse or death in the subgroups of free 25(OH)D levels lower (**A**) and higher (**B**) than median levels (5.70 pg/mL) before starting supplementation. 25(OH)D: 25-hydroxyvitamin D; CI: confidence interval; 5 y RFS: 5 year relapse-free survival.

**Figure 5 cancers-12-00347-f005:**
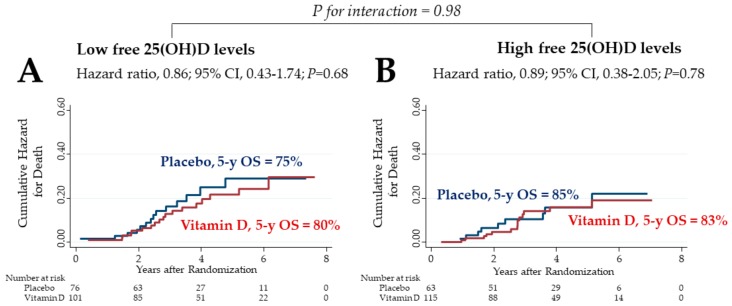
Nelson–Aalen cumulative hazard curves for death in the subgroups of free 25(OH)D levels lower (**A**) and higher (**B**) than median (5.70 pg/mL) before starting supplementation. 25(OH)D: 25-hydroxyvitamin D; CI: confidence interval; 5 y OS: 5 year overall survival.

**Figure 6 cancers-12-00347-f006:**
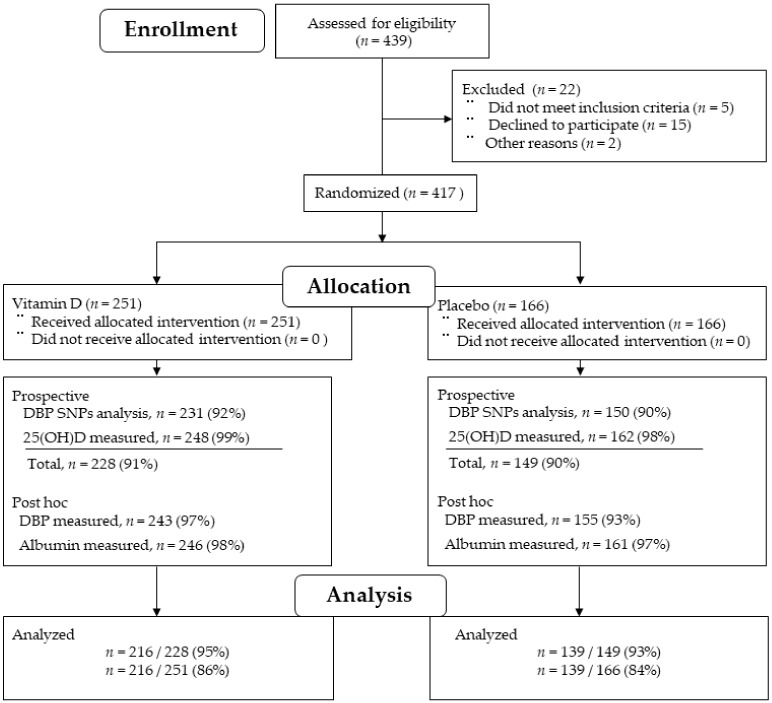
Flow diagram of patients. DBP: vitamin-D-binding protein; SNP: single nucleotide polymorphism; 25(OH)D: 25-hydroxyvitamin D.

**Table 1 cancers-12-00347-t001:** Participants’ characteristics and serum 25(OH)D-related variables used in this post hoc analysis.

Characteristics	No. (%) of Participants	
	Vitamin D *n* = 216	Placebo *n* = 139
Male, *n* (%) *	152 (70)	84 (60)
Age—years, median (IQR)	67 (61–76)	64 (58–70)
Site of cancer, *n* (%) *		
Esophagus	20 (9)	12 (9)
Stomach	94 (44)	64 (46)
Small bowel	1 (0.5)	1 (0.7)
Colorectal	101 (47)	62 (45)
Stage, *n* (%) *		
I	103 (48)	59 (42)
II	50 (23)	37 (27)
III	63 (29)	43 (31)
Pathology, *n* (%) *		
Adenocarcinoma	194 (90)	125 (90)
Squamous cell carcinoma	20 (9.3)	12 (8.6)
Neuroendocrine tumor	2 (0.9)	2 (1.4)
Total 25(OH)D—ng/mL, median (IQR)	21.5 (16.5–27.5)	21 (15–27)
Albumin—g/dL, median (IQR)	3.5 (3.1–3.9)	3.5 (3.1–4.0)
Vitamin-D-binding protein—µg/mL, median (IQR)	255 (210–303)	263 (223–316)
SNPs, *n* (%) *		
rs7041		
GG	10 (4.6)	9 (6.5)
GT	82 (38)	55 (40)
TT	124 (57)	75 (54)
rs4588		
CC	106 (49)	78 (56)
CA	87 (40)	52 (37)
AA	23 (11)	9 (6.5)
Free 25(OH)D—ng/mL, median (IQR)	6.00 (3.92–9.11)	5.43 (3.47–7.11)
Bioavailable 25(OH)D—ng/mL, median (IQR)	1.80 (1.23–2.80)	1.64 (1.10–2.29)

* Percentages may not equal 100% because of rounding. Interquartile range: IQR; 25(OH)D: 25-hydroxyvitamin D; SNP: single nucleotide polymorphism.

**Table 2 cancers-12-00347-t002:** Allele frequencies of DBP and its effect on total 25(OH)D levels.

SNP Allele				Association with Total 25(OH)D Levels Per Variant Allele Copy		
	Reference	Variant	Frequency of variant	Coefficient	95% CI	*p*-value
rs7041	G (Glu)	T (Asp)	0.76	−1.38	−2.73 to −0.02	0.047
rs4588	C (Thr)	A (Lys)	0.29	−1.33	−2.56 to −0.11	0.03

VDP: vitamin D-binding protein; 25(OH)D: 25-hydroxyvitamin D; CI: confidence interval.

**Table 3 cancers-12-00347-t003:** Safety outcomes.

Outcomes	No (%) of Participants			
	Low Bioavailable 25(OH)D Levels, *n* = 177		High Bioavailable 25(OH)D Levels, *n* = 178	
	Vitamin D *n* = 101	Placebo *n* = 76	Vitamin D *n* = 115	Placebo *n* = 63
Fracture, *n* (%) *	1 (1.0)	5 (6.6) *	2 (1.7)	1 (1.6)
Urinary stones, *n* (%)	0 (0.0)	1 (1.3)	2 (1.7)	3 (4.8)
Severe adverse events, ^a^ *n* (%)	8 (7.9)	9 (12)	11 (9.6)	3 (4.8)
Cancer de novo, ^b^ *n* (%)	4 (4.0)	4 (5.3)	12 (10)	3 (4.8)

* *p* = 0.04 between the vitamin D and placebo group. ^a^ Adverse events that resulted in admission. ^b^ Cancer that appeared de novo in organs other than the site of the primary cancer after starting supplementation.

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
