# Peer review of "Effect of Vitamin D Supplementation on Survival of Digestive Tract Cancer Patients with Low Bioavailable 25-Hydroxyvitamin D Levels: A Post Hoc Analysis of the AMATERASU Randomized Clinical Trial"

_cancers, 2020, doi:10.3390/cancers12020347_

Round 1

Reviewer 1 Report

This is the second post-hoc analysis of the AMATERASU randomized
clinical trial (an additional post-hoc analysis of the same trial has already been published in Nutrients 2 months ago by some of the authors of the current study).

This is a well writen manuscript.

However, the meta-analysis presented here suffers form several limitations (aknowledged by the authors themselves, ln217-236), which I feel are sufficinet to prevent publication in Cancers.

Reviewer 2 Report

This post hoc analysis by Urashima M et al, used residual serum samples and clinical data from the AMATERASU trial. The aim of the study was to explore whether vitamin D supplementation improved survival in a subgroup of patients with lower than median bioavailable 25(OH)D levels. The paper is straightforward, well written, and concise. Definitely deserves to be published and is a valuable contribution to the “cancers” journal. Some minor flaws need to be addressed before publication.

Minor points:

[1] Introduction, Lines 35-41:

A  few more words about the synthesis of vitamin D would be added here. 1,25[OH]2D is metabolized by the enzyme CYP24A1 that limits calcitriol actions via catabolism. Renal CYP27B1 gene expression is regulated and activated by the parathyroid hormone. After its synthesis, 1,25[OH]2D is released into the serum and can act beyond bone, on the intestine, and kidney to regulate calcium metabolism. CYP27B1 and CYP24A1 are found in numerous tissues throughout the body including the skin, colon, pancreas, liver, brain and placenta allowing for 1,25[OH]2D synthesis and degradation.

Relevant reference: Picotto G, et al. Molecular aspects of vitamin D anticancer activity. Cancer Invest. 2012 Oct;30(8):604-14.

[2] Discussion, Line 216:

Before the statement of study’s limitations, please make a comment about the importance of monitoring serum vitamin D levels closely when used for treatment, as there can be developed other side effects related to overload of vitamin D. Identification of the optimal form and dosage of vitamin D supplementation that may generate an effective protection against cancer warrants further investigation.

Relevant references:

(a) Stolzenberg-Solomon RZ, et al. Circulating 25-hydroxyvitamin D and risk of pancreatic cancer: Cohort Consortium Vitamin D Pooling Project of Rarer Cancers. Am J Epidemiol. 2010 Jul 1;172(1):81-93.

(b) Melamed ML, et al. Vitamin D and cardiovascular disease and cancer: not too much and not too little? The need for clinical trials. Womens Health (Lond). 2011 Jul;7(4):419-24.

Reviewer 3 Report

In present paper, “Effect of Vitamin D Supplementation on Survival of Digestive Tract Cancer Patients with Low Bioavailable 25-hydroxyvitamin D levels: A Post Hoc Analysis of the AMATERASU Randomized Clinical Trial”, the authors aimed to assess the impact of vitamin D supplementation,  bioavailable 25(OH)D, on overall survival and relapse-free survival (RFS) in patients with cancer of the digestive tract, using data obtained from a randomized clinical trial performed by the group. The study has some limitations, which were mentioned by the authors, but it showed interesting results. However, some questions and considerations need to be reviewed and answered by the authors.

Minor issues:

1- The first paragraph of item 2.1 of the results (lines 68-71) and figure 1 should be part of the methodology.

2- In the results, the authors said that “Median total 25 (OH) D levels increased significantly in the vitamin D group but did not change significantly in the placebo group (data previously reported) (lines 102-103)”. In this previously reported (Urashima et al., 2019), patients were classified into subgroups according to low, middle or high serum levels of 25 (OH) D at baseline.  However, data on the analysis of the total 25(OH D levels were not shown.

3- Add in the methodology that bioavailable 25(OH)D and free 25 (OH)D groups were stratified by the median in analysis of relapse-free survival (RFS) and overall survival.

4- What is the p value in analyzes between placebo and vitamin D in each graph of figures 3 (A and B); 4 (A and B); 5 (A and B) and 6 (A and B)? Add in the text, please.

5- I didn't understand the sentence: “We hypothesized that the mechanism of increase in DBP occurred through surgical stress and reduced total 25 (OH) D levels, which subsequently increased DBP levels through a negative feedback loop; the increased DBP helped total 25 (OH) D levels recover ”(lines 197-200). Please, rewrite.

6-Figure 5 –B: Low free 25(OH)D levels instead of High free 25(OH)D levels.

7-Material and Methods item is confused, I suggest rewrite this in subitems.

8- There is no reference on line 212.

Major  issues:

The abstract lacks an initial sentence that objectively justifies the relationship between cancer cells and vitamin D. Another important issue was the mention of two hypotheses suggested by the authors - bioavailable 25 (OH) D as a good marker for detecting vitamin D insufficiency, and the improved survival of cancer patients with low bioavailable levels of 25 (OH) D by supplementation with vitamin D.- but only one is explored as the goal of the study. In this sense, I would like to read more about the use of bioavailable 25 (OH) D as a biomarker of vitamin D deficiency compared to that currently used in other studies. The authors performed analysis using different tumor sites and different histopathologic subtypes (adenocarcinoma and squamous cell carcinoma). Why did they not do analysis of stomach and colorectal cancer separately? They are the tumor types that have a greater number of patients per group. Why did they use neuroendocrine tumors and esophageal cancers (EC) (EC are squamous cell carcinoma) for analysis? These tumors are very different from adenocarcinomas.

3- Although the authors mentioned in the previous article that “(…) these SNPs are missense mutations that influence serum levels of 25(OH)D”.

It was not clear the phenotypic consequence of the polymorphisms of DBPs. I suggest adding this explanation in the text. Are these polymorphisms in Hardy Weinberg equilibrium? This information must appear in the text.

4- The authors consider the evaluation of bioavailable 25 (OH) D as an important variable in the patients groups clustering.  Therefore, I suggest the authors to discuss more about serum free and bio-available 25-hydroxyvitamin D and the uptake of vitamin D by cancer cells. 

Reviewer 4 Report

This manuscript describes the results of a post hoc analysis of a randomized double blind placebo-controlled trial of postoperative oral vitamin D3 supplementation in patients with stage I to III digestive tract cancer. The authors aimed to examine the hypothesis that vitamin D supplementation may be effective in improving relapse free survival among digestive tract cancer patients with low bioavailable 25(OH)D levels. The manuscript was very well prepared. There are a few minor comments that may need to be addressed.

Abstract, lines 17-18, the sentence doesn’t seem to make sense. Lines 69-71, the explanation of the numbers of subjects in both the text and in the Figure 1 are not clear. Are the two numbers listed in Figure 1 under “Prospective” those subjects whose measurements were completed during the main study? Then, the two additional measurements under “Post hoc” were completed only for the post hoc study? Please clarify. Figure 5B, there appears to be a typo in the title. Table 3, is the difference in fracture % between the two groups of low bioavailable 25(OH)D levels the only one statistically significant?

Round 2

Reviewer 1 Report

This is the revised version of a previously submitted manuscript. I insist on my original decision, but of course the work can be considered for publication as long as the rest of reviewers are covered by the authors' replies to their concerns raised during the previous round of the reviewing process.